# Multiobjective Optimization of the Energy Efficiency and the Steam Flow in a Bagasse Boiler

Ducardo L. Molina [1], Juan Ricardo Vidal Medina [1], Alexis Sagastume Gutiérrez [2,*], Juan J. Cabello Eras [3], Jesús A. Lopez [1], Simón Hincapie [1] and Enrique C. Quispe [1]

[1] Grupo de Investigación en Energías (GIEN), Faculty of Engineering, Universidad Autónoma de Occidente, Cll 25#115-85, Cali 760030, Colombia
[2] Department of Energy, Universidad de la Costa, Calle 58 No. 55-66, Barranquilla 080002, Colombia
[3] Department of Mechanical Engineering, Universidad de Cordoba, Cra. 6 No. 77-305, Cordoba 230002, Colombia
[*] Correspondence: asagastu1@cuc.edu.co

**Abstract:** Renewable energy and energy efficiency are essential for a transition to cleaner and sustainable energy. Photovoltaic and wind turbine systems introduce operation, control, protection, and planning issues, particularly affecting frequency stability in the grid. In contrast to more widespread wind turbines and photovoltaic systems, biomass based electricity systems are more stable with no negative impacts on the grid stability. The efficiency of bagasse boilers is essential to guaranteeing adequate economic profit and environmental performance in sugar plants. To realize universal access to affordable, reliable, and modern energy services by 2030 (SDG 7), the use of renewable energy sources in energy mixing and energy efficiency must increase globally. Sugar plants include cogeneration systems to provide heat and electricity to the process and frequently sell an electricity surplus to the grid, which depends on their energy efficiency. Boilers are an essential component of cogeneration systems in sugar plants, and their efficiency is crucial to guarantee electricity surplus. Therefore, this study assessed a bagasse boiler to optimize its operational efficiency. To this end, the exergy assessment and multiobjective optimization based on a genetic algorithm are used. The results show that the exergy efficiency of the boiler improved by 0.8% with the optimization, reducing bagasse consumption by 23 t/d.

**Keywords:** water-tube boilers; cogeneration; energy efficiency; exergy efficiency; bagasse

## 1. Introduction

The industrial sector faces many economic and environmental challenges, including increasing energy and raw materials costs, global warming, and resource depletion. Thus, new strategies to achieve sustainability and competitiveness, motivated to reduce energy costs while reducing GHG emissions, are under development in the industry [1]. A way to reduce GHG emissions in the industry is to improve energy efficiency, which is projected to be critical in meeting the GHG emissions reduction targets defined by 2050 [2]. Realizing sustainable development goals (SDGs) is essential for developing countries to reach the development of industrialized countries while balancing the use of natural resources with socioeconomic development [3]. Environmental sustainability is achievable, among other factors, by reducing the negative environmental consequences of fossil fuel-based electricity generation with renewable energy sources [4]. Notably, bagasse boilers have gained increased attention since the 1970s energy crisis [5].

Although energy consumption is a cornerstone in many countries' long-term development, it is increasingly evident that renewable energy and energy efficiency are a cornerstone to addressing the climate crisis while meeting sustainable development goals [6]. Primarily based on fossil fuels, electricity is the energy carrier most widely used globally [7]. Researchers globally acknowledge that further use of fossil fuels continuously

raises pollution levels [8]. However, the transition in the electric systems points to replacing fossil fuel based synchronous electricity generation systems with inverter interfaced renewable based energy sources such as photovoltaic panels, wind turbines, and batteries [9]. These systems are decoupled from the grid and provide no inertial support, which results in a low inertial system [10] and has caused blackouts worldwide [11]. Lower grid inertia results in grid issues in system operation, control, protection, and planning, predominantly presenting issues with frequency stability [11,12]. In contrast, bagasse-based electricity generation technology in sugar plants is similar to fossil-fuels based systems and has no negative impact on the grid inertia. Consequently, bagasse-based electricity is a sound alternative to increase clean energy availability. Moreover, energy efficiency in sugar plants can further increase the available clean electricity in the grid.

Steam production in boilers is highly irreversible, driving the rising concern for the correct operation of industrial boilers. Different studies have aimed to improve the energy performance of boilers using trial-and-error approaches or based on complex mathematical models. Some mathematical models include the heat exchange in the boiler surfaces simulated by integrating combustion and hydrodynamic models [13]. Other mathematical models were developed to describe the pressure drop in superheater heat exchangers [14,15]. Moreover, different computational tools have been used to analyze the phase change process and the pressure drops in boilers [16]. A hybrid mathematical model was used for the dynamic operation of a boiler during transient processes such as hot and cold starts [17]. Combustion irreversibilities, chemical energy transformation, and heat transfer between combustion gases and steam have been simulated using mathematical models [18]. Validating the mathematical model used in boilers needs data from different partial loads from operations [19].

Different studies discussed the energy performance in the sugar industry. An experimental energy performance optimization was developed in three RETAL boilers in two sugar plants in Cuba [5]. The optimization was based on the indirect efficiency method and a minimum total cost function. This experimental approach proved to be resource and time-consuming. A different study combined the indirect efficiency with a non-linear optimization using fmicon function in Matlab to optimize the fuel consumption in a bagasse boiler [20]. Another study assessed a bagasse boiler using the second law of thermodynamics to quantify improvements in the boiler's performance for increased steam temperatures and pressures [21]. The exergy balance in an Indian sugar plant shows that combustion and heat transfer account for most irreversibilities in bagasse boilers operating at an exergy efficiency of 25% [22]. Likewise, the electricity output of a sugar plant cogeneration system was optimized using an exergy-based thermoeconomic approach [23]. The exergy assessment of a bagasse boiler in Colombia identified the need for technological upgrades to improve the exergy efficiency from 25.8% to 27% with a reheating cycle [24]. An integrated multi-criteria tool to evaluate the sustainability of bagasse burning in a Mexican sugar mill reduced GHG emissions by 55% and particulate matter emissions by 58% [25]. Using multiobjective optimization based on the simulation of the steam heaters with a focus on steam generation permits the improvement of the boilers' control loops [26]. A different study assessed the competing uses of bagasse for energy and non-energy applications in 31 sugar plants in India with three objective functions: NPV maximization, GHG emissions minimization, and water footprint minimization [27]. The results show that using bagasse for electricity production optimizes the NPV and GHG emissions. A comparative analysis of four technologies in a Cuban sugar plant shows the potential of technological improvements to upgrade energy efficiency and sugar yields while reducing GHG emissions and economic costs [28].

It is challenging to simultaneously optimize maximum steam flow and power output in boilers to obtain optimum efficiency. Thus, it is necessary to balance output and efficiency in boilers. Consequently, a multiobjective optimization approach considering different cost functions is indicated to optimize these parameters simultaneously. Meta-heuristic techniques are widely used in complex optimization problems, with different algorithms

available [29]. These algorithms have been used in the multiobjective optimization of Stirling engines [30,31]. Moreover, exergy analysis is a practical approach for identifying and quantifying losses and irreversibilities in boilers [32]. The exergy approach permits highlighting inefficiencies in the different components of power systems [33]. Therefore, the main question is as follows: How does one define the best operating point in a bagasse boiler combining multiobjective optimization and exergy assessment? Thus, this work aims to implement a multiobjective optimization of a bagasse boiler's efficiency and steam flows.

## 2. Materials and Methods

This section describes the Pareto front optimization approach used in this study and its implementation in the ModeFRONTIER software. Additionally, it presents the main steam boiler characteristics. Furthermore, it includes a description of the mathematical model of the steam boiler used in the optimization.

### 2.1. Steam Boiler

This study assesses a cogenerating sugarcane bagasse steam boiler of 34 MW. The boiler consists of a two drum water tube system with natural circulation. Furthermore, the boiler can operate with either bagasse (Lower Heating Value (LHV) of 8839 kJ/kg), coal (LHV of 23,818 kJ/kg), or a mix of both fuels. On average, bagasse accounts for some 93% of the fuel energy, while coal contributes to the remaining 7% during regular operation. Table 1 shows the technical characteristics of the steam boiler.

**Table 1.** Characteristics of the boiler.

| Features | Unit | Value |
| --- | --- | --- |
| Primary fuel | - | Bagasse |
| Secondary fuel | - | Coal |
| Grate | - | Traveling grate with frontal discharge |
| Maximum fuel flow (100% bagasse) | t/h | 181 159 (100% Carbón) |
| Nominal pressure * | MPa | 7.991 |
| Live steam temperature | °C | 510 |
| Feed water temperature | °C | 125 |
| Thermal efficiency (100% bagasse) | % | 66.71 |
| Thermal efficiency (100% coal) | % | 83.17 |
| Steam production (100% bagasse) | $kg_s/kg_f$ | 2.01 |
| Steam production (100% coal) | $kg_s/kg_f$ | 6.67 |

* Nominal pressure refers to the maximum steam outlet pressure of the boiler.

The sugar plant produces enough bagasse to support its energy demand. However, an agreement with a paper plant establishes an exchange of some bagasse for coal. The paper plant provides coal to the sugar plant in exchange for bagasse to support paper production.

The boiler is considered a system formed by different subsystems for assessment. Figure 1 describes the systems and flows interaction within the steam boiler.

In the boiler, combustion gases flow into the steam production system (i.e., the steam and feedwater drums and the water tubes) to produce saturated steam. Afterward, the gases flow to the superheaters, producing high pressure and temperature superheated steam. The gases flow to the overfire air heater and to the air preheater to preheat combustion air. Finally, combustion gases flow to the economizer to preheat the input water to the boiler. The steam temperature at the exit of superheater 2 is one critical operational parameter in the boiler, affecting process safety, efficiency, and emissions.

Table 2 shows the values of the operational parameters of the boiler for the different flows depicted in Figure 1.

The boiler produces high temperature superheated steam at 513 °C and 6.7 MPa to support the sugar plant cogeneration system.

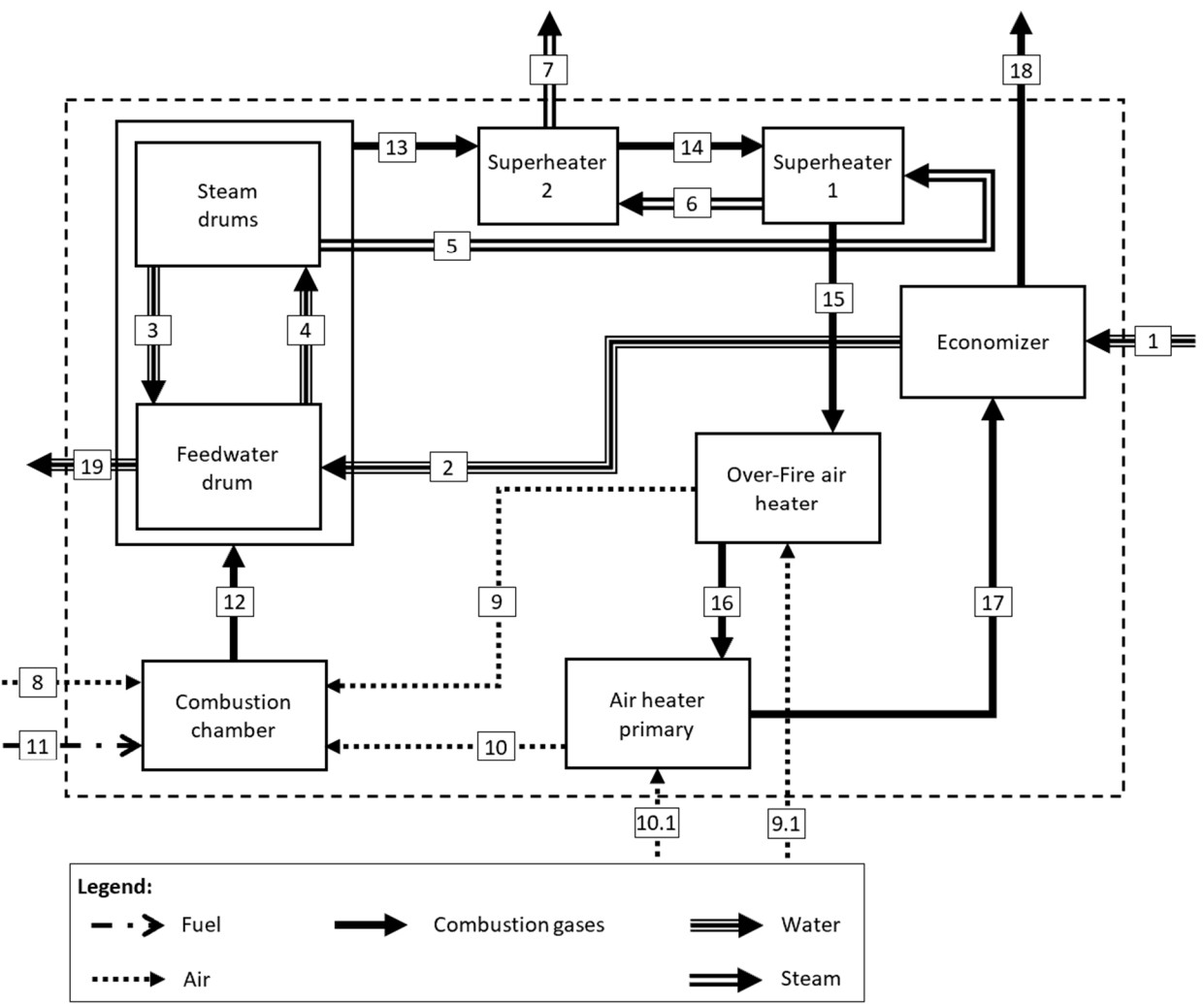

**Figure 1.** Steam boiler.

**Table 2.** Operational parameters of the regular boiler operation.

| Fluid | Flow | Flow (t/h) | Temperature (°C) | Pressure (MPa) |
|---|---|---|---|---|
| Feedwater | 1 | 166.5 | 116.5 | 7.72 |
| | 2 | 166.5 | 185 | 7.72 |
| Saturated steam | 5 | 164.95 | 290 | 7.40 |
| Superheated steam | 6 | 164.95 | 400 | 7.10 |
| | 7 | 164.95 | 513 | 6.71 |
| Pneumatic air | 8 | 21.18 | - | 0.030 |
| Over-fire air | 9 | 77.8 | 384 | - |
| Primary air | 10 | 177.2 | 243 | - |
| Fuel | 11 | 80.212 | 30 | - |

**Table 2.** *Cont.*

| Fluid | Flow | Flow (t/h) | Temperature (°C) | Pressure (MPa) |
|---|---|---|---|---|
| | 12 | 388.25 | 1442 | - |
| | 13 | 388.25 | 755 | - |
| | 14 | 388.25 | 527 | - |
| Combustion gases | 15 | 388.25 | 443 | - |
| | 16 | 388.25 | 374 | - |
| | 17 | 388.25 | 277 | - |
| | 18 | 388.25 | 180 | - |
| Continuous purge | 19 | 1.55 | 290 | 7.40 |
| Air | 9.1 | 77.8 | 30 | 0.101 |
| | 10.1 | 177.2 | 30 | 0.101 |

*2.2. Pareto Front Multiobjective Optimization*

Evolutionary optimization for multiple objective optimization problems has been increasingly discussed in engineering problems [34] and has gained a preference towards sustainability [35]. Evolutionary optimizations identify a population of possible solutions approximating the Pareto front, defined as a set of trade-off solutions between given criteria [36]. Multiobjective optimization methods aim to balance two or more conflicting objectives to achieve optimal decisions [37]. The most important feature of these methods is that more than one candidate solution is obtained, identifying a set of optimum solutions rather than a single optimum solution [38].

The multiobjective optimization problem aiming at maximizing or minimizing the m objective functions evaluated at different values of the decision variables vector $x$ with $n$ decision variables [39]:

$$min \ [f_1(x) \ f_2(x) \ \ldots f_m(x)] \tag{1}$$

where:

$$x = (x_1, x_{2,\ldots}, x_{n,}) \in X \tag{2}$$

with the constraints:

$$g(x) \leq 0 \tag{3}$$

$$h(x) = 0 \tag{4}$$

$$x_l \leq x \leq x_u \tag{5}$$

where $X \subseteq \Re n$ is the n-dimensional decision space and functions $g(x)$ and $h(x)$ are the constraint functions. The function value f is evaluated at point $x_i$ ($f_1(x_i), f_2(x_i),\ldots, f_n(x_i),$).

The set of Pareto optimal solutions obtained is referred to as *Pareto set*, and its image is referred to as the Pareto front [40,41].

In multiobjective optimization, a solution is optimal if, assuming minimization, there is no feasible vector $x$ that would decrease some criterion without causing a simultaneous increase in at least one other criterion [29]. Otherwise, assuming maximization, the solution is optimal if there were no feasible vector $x$ that would increase some criterion without causing a simultaneous decrease in at least one other criterion. This set of solutions (located on the Pareto front) is known as nondominated solutions [29,42]. All other solutions outside the Pareto front are dominated solutions [42]. Every solution associated with a point on the Pareto front is a vector whose components represent trade-offs in the decision space [29,42]. Therefore, without additional information, Pareto optimal solutions are considered equally good [43,44]. Selecting an optimal solution thus depends on the trade-

offs between objective functions. For example, in a boiler is possible to optimize thermal efficiency and superheated steam flow; the selection of an optimal nondominated solution depends on the superheated steam demand. Therefore, the optimal solution, in this case, guarantees the highest efficiency for a given steam flow.

Evolutionary algorithms are an efficient approach to exploring the Pareto front, and the Elitist Non-Dominated Sorting Genetic Algorithm (NSGA-II) is one of the most used algorithms for multiobjective optimization [45]. NSGA-II has been increasingly implemented for different multiobjective optimization problems in engineering, performing better than other algorithms [46]. The superior performance f NSGA-II is based on the mutation operator, population distribution precluding local optimum points entrapment, good crowding distance operator performance during selection, and well-preserved population diversity through different generations [47]. It is suggested that the proposed NSGA-II be used to optimize multiobjective system operation with more reservoirs.

The NSGA-II algorithm and The ModeFRONTIER software are used to develop the multiobjective optimization.

One significant challenge to address when obtaining a Pareto front is the decision-making process to select the solution to be implemented in the actual process. Various approaches have been discussed to select the optimal operational configuration within the Pareto front. Ferreira et al. [48] suggested a method with a weighted stress function to incorporate the user's preferences in the decision. This approach enables the identification of the optimal region within the Pareto frontier that aligns with the user's preferences. A different approach [49] considers the standard error of the predictions for the responses during the solution selection process. Moreover, the R-Method involves ranking the objectives based on their importance within the optimization problem and ranking the alternative solutions based on their corresponding objective data [50]. In this case, the ranks assigned to the objectives, and the alternative solutions for each objective are then converted into appropriate weights. Next, these weights are used to compute the final composite scores for the alternative solutions, and the ranking of solutions is determined based on these scores. This study implements the max-min strategy [51] to select the optimal operational configuration in the Pareto Front. The solution in this case is selected as the closest one that satisfies the operational constraints in the actual process. This approach involves the formulation of normalized vectors for each objective function, based on the following equation [52]:

$$\mathrm{MF_k} = \frac{\mathrm{F_k^{max}} - \mathrm{F_k}}{\mathrm{F_k^{max}} - \mathrm{F_k^{min}}} \tag{6}$$

where $\mathrm{F_k^{max}}$ is the maximum value of the kth objective function, $\mathrm{F_k^{min}}$ is the minimum value of the kth objective function, and $\mathrm{F_k}$ is the value of the kth objective function.

The optimal value for this study is derived from the highest value of the new vector generated with the max-min strategy. This strategy permits the identification of the best solutions within the Pareto front.

### 2.3. Mathematical Model

A mathematical model based on mass, energy, and exergy balances was developed for each subsystem to simulate the boiler performance. The model used in this study combines the first and second principles of thermodynamics and the concept of exergy in the balances.

The exergy of a flow, which depends on the reference temperature and pressure, is calculated as in [53]:

$$\sum \left(1 - \frac{T_0}{T_k}\right) \cdot \dot{Q}_k - \left[\dot{W} - P_0 \cdot \frac{dV_{CV}}{dt}\right] + \sum\nolimits_{in} \dot{m}_i \cdot x_i - \sum\nolimits_{out} \dot{m}_j \cdot x_j - \dot{X}_D = \frac{dX_{CV}}{dt} \tag{7}$$

where:

$$x = (h - h_o) - T_o \cdot (s - s_o) \tag{8}$$

For ideal gases such air and combustion gases, this equation is simplified to:

$$x_g = c_{p_g} \cdot (T_g - T_o) - T_o \cdot \left[ c_{p_g} \cdot Ln\left(\frac{T_g}{T_o}\right) - R_g \cdot Ln\left(\frac{P_g}{P_o}\right) \right] \tag{9}$$

where the exergy destroyed is calculated with the Gouy-Stodola relation [53]:

$$\dot{X}_D = \dot{S}_{gen} \cdot T_o \tag{10}$$

The entropy generation is calculated as follows:

$$\dot{S}_{gen} = \frac{dS}{dt} - \sum_{k=0}^{n} \frac{\dot{Q}_k}{T_k} - \sum_{in} \dot{m} \cdot s + \sum_{out} \dot{m} \cdot s \geq 0 \tag{11}$$

The exergy of the fuel is calculated as follows:

$$\dot{X}_f = \dot{m}_f \cdot (\varphi \cdot LHV) \tag{12}$$

where $\varphi$ is the ratio between the chemical exergy and the lower heating value of industrial fuels. For solid fuels, $\varphi$ is a function of hydrogen ($H$), oxygen ($O$), nitrogen ($N$), and carbon ($C$) [54]:

$$\varphi = 1.0437 + 0.1882 \cdot \frac{H}{C} + 0.0610 \cdot \frac{O}{C} + 0.0404 \cdot \frac{N}{C} \tag{13}$$

The energy and exergy balance of the steam boiler is shown in Table 3.

**Table 3.** Energy and exergy balance of the steam boiler.

| System | | Equation | No. |
|---|---|---|---|
| Combustion chamber | A | $\dot{m}_8 \cdot h_8 + \dot{m}_{11} \cdot LHV = \dot{m}_{12} \cdot h_{12}$ | (14) |
| | | $X_8 + X_{11} - \dot{X}_{D_I} = X_{12}$ | (15) |
| Steam production system | B | $\dot{m}_2 \cdot h_2 + \dot{m}_{12} \cdot h_{12} = \dot{m}_5 \cdot h_5 + \dot{m}_{13} \cdot h_{13} + \dot{m}_{19} \cdot h_{19}$ | (16) |
| | | $X_2 + X_{12} - \dot{X}_{D_{II}} = X_5 + X_{13} + X_{19}$ | (17) |
| Superheater I | C | $\dot{m}_5 \cdot h_5 + \dot{m}_{14} \cdot h_{14} = \dot{m}_6 \cdot h_6 + \dot{m}_{15} \cdot h_{15}$ | (18) |
| | | $X_5 + X_{14} - \dot{X}_{D_{III}} = X_6 + X_{15}$ | (19) |
| Superheater II | D | $\dot{m}_6 \cdot h_6 + \dot{m}_{13} \cdot h_{13} = \dot{m}_7 \cdot h_7 + \dot{m}_{14} \cdot h_{14}$ | (20) |
| | | $X_6 + X_{13} - \dot{X}_{D_{IV}} = X_7 + X_{14}$ | (21) |
| Over fire air heater | E | $\dot{m}_{9.1} \cdot h_{9.1} + \dot{m}_{15} \cdot h_{15} = \dot{m}_9 \cdot h_9 + \dot{m}_{16} \cdot h_{16}$ | (22) |
| | | $X_{9.1} + X_{15} - \dot{X}_{D_V} = X_9 + X_{16}$ | (23) |
| Primary air heater | F | $\dot{m}_{10.1} \cdot h_{10.1} + \dot{m}_{16} \cdot h_{16} = \dot{m}_{10} \cdot h_{10} + \dot{m}_{17} \cdot h_{17}$ | (24) |
| | | $X_{10.1} + X_{16} - \dot{X}_{D_{VI}} = X_{10} + X_{17}$ | (25) |
| Economizer | G | $\dot{m}_1 \cdot h_1 + \dot{m}_{17} \cdot h_{17} = \dot{m}_2 \cdot h_2 + \dot{m}_{18} \cdot h_{18}$ | (26) |
| | | $X_1 + X_{17} - \dot{X}_{D_{VII}} = X_2 + X_{18}$ | (27) |

Based on the exergy balance is possible to assess the boiler efficiency. According to [21], the exergy efficiency of a boiler ($\eta_{boiler}$) can be determined with two different approaches:

1. The ratio of exergy gained by water between the input and output of the boiler to the fuel exergy:

$$\eta_{boiler} = \frac{\dot{X}_{SH}}{\dot{X}_{in}} \tag{28}$$

2. The ratio of the exergy destroyed and lost in each subsystem to the fuel exergy (this approach provides more information on the boiler's performance) [55]:

$$\eta_{boiler} = 100 - \frac{\sum \dot{X}_D + \sum \dot{X}_{loss}}{\dot{X}_{in}} \tag{29}$$

Efficiency approach one shows the boiler efficiency, providing no information on the inefficiencies of the different subsystems (e.g., superheaters). Moreover, although more complex, the second approach permits identifying and quantifying the sources of exergy destruction; thus, it is adequate for detecting and locating inefficiencies. Therefore, this study used the second approach to calculate the exergy efficiency of the boiler. The mathematical model is programmed in MATLAB.

The objective functions of the multiobjective optimization model are the exergy efficiency defined by the ratio of the exergy destroyed and lost in each subsystem (Equation (28)) and the superheated steam production (Equation (20)) in the boiler $\left( \eta_{boiler}; \dot{m}_7 \right)$.

$$max[f_1(x)f_2(x)] \tag{30}$$

$$f_1(x) = \eta_{boiler} = 100 - \frac{\sum \dot{X}_D + \sum \dot{X}_{loss}}{\dot{X}_{in}} \tag{31}$$

$$f_2(x) = \dot{m}_7 = \frac{\dot{m}_6 \cdot x_6 + \dot{m}_{13} \cdot x_{13} - \dot{X}_{D_{IV}} - \dot{m}_{14} \cdot x_{14}}{x_7} \tag{32}$$

The constraints of each operational variable used in the steam boiler model and the optimization, defined by the operating range, are shown in Table 4. These variables coincide with the decision variables of the multiobjective optimization.

**Table 4.** Operating parameters constraints in the optimization model.

| Parameter | Unit | ModeFRONTIER Nomenclature | Range Considered |
|---|---|---|---|
| Steam temperature | °C | $T_7$ | 505–515 |
| Steam pressure | kPa | $P_7$ | 6541–6580 |
| Feedwater temperature (economizer input) | °C | $T_1$ | 120–130 |
| Feedwater flow (economizer output) | t/h | $F_1$ | 175–190 |
| Feedwater pressure | kPa | $P_1$ | 6669–6865 |
| Primary airflow | m$^3$/s | $F_{10}$ | 20–50 |
| Primary airflow temperature | °C | $T_{10}$ | 190–210 |
| Overfire airflow | m$^3$/s | $F_9$ | 7–20 |
| Overfire airflow temperature | °C | $T_9$ | 280–315 |
| Pneumatic airflow | m$^3$/s | $F_8$ | 5–16 |
| Pneumatic airflow temperature | °C | $T_8$ | 30 |
| Combustion gas temperature (Chimney output) | °C | $T_{18}$ | 165–175 |
| Steam production system temperature | °C | $T_{12}$ | 780–943 |
| Combustion gas temperature (Overfire air heater) | °C | $T_{15}$ | 377–450 |
| Combustion gas temperature (Primary air heater) | °C | $T_{16}$ | 280–370 |
| Combustion gas temperature (Economizer) | °C | $T_{17}$ | 225–280 |

The multiobjective function is optimized using the NSGA-II and MATLAB integrated into ModeFRONTIER, as shown in Figure 2. The optimization model includes the mathematical model and the objective functions.

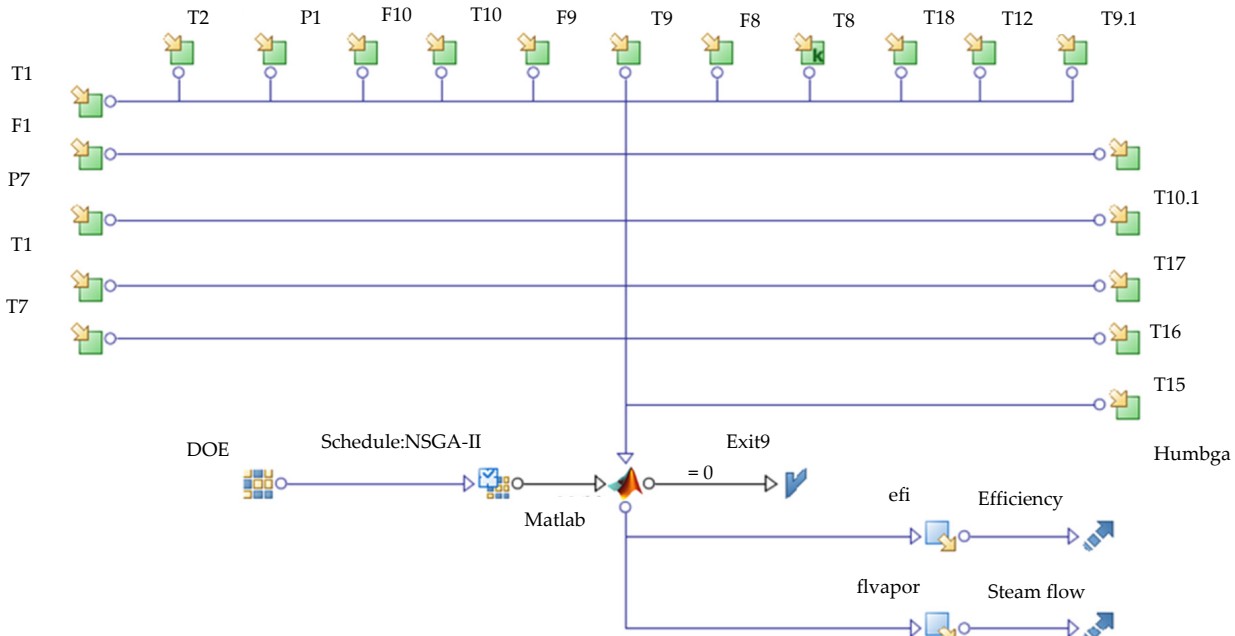

**Figure 2.** ModeFRONTIER workspace.

The workspace shown in Figure 2 presents the input variables used in the model, which is combined with the NSGA-II optimization method within the ModeFRONTIER. The evolutionary algorithm compares the results obtained from the model for different inputs to define the Pareto front. The design of experiments (DOE) model was developed using the Sobol method, which is suitable for medium to large samples [56]. The NSGA-II selects the best operating configuration in the Pareto front using the max-min technique, which guarantees the diversity of subsets by maximizing the minimum distance between the selected elements [51,52].

When setting genetic algorithm parameters, it is recommended to increase the population size rather than the generations [57]. Based on the model characteristics, the optimization was developed using the genetic algorithm parameters shown in Table 5.

**Table 5.** Genetic algorithm parameters.

| Parameter | Value |
|---|---|
| Population size | 500 |
| Number of generations | 50 |
| Probability of directional | 0.5 |
| Probability of selection | 0.05 |
| Probability of mutation | 0.1 |

Several iterations of the algorithm were assessed by comparing computational demand and the Pareto front to define the value of these parameters. During the tests, the population increased to 500 and the generations increased up to 50. Further increments did not improve the algorithm's performance but increased the computational demand.

## 3. Results and Discussion

This section presents the implementation of steam boiler optimization. The mathematical model described in Section 2 was validated using operating parameters measured during the regular operation of the steam boiler. Once the model is validated, it is combined with the optimization approach. Finally, the results from optimization are implemented in the steam boiler operation.

### 3.1. Model Validation

The temperature values of different flows measured during the regular operation of the boiler for 22 days were compared with the results of the mathematical model. Figure 3 shows the comparison of the results.

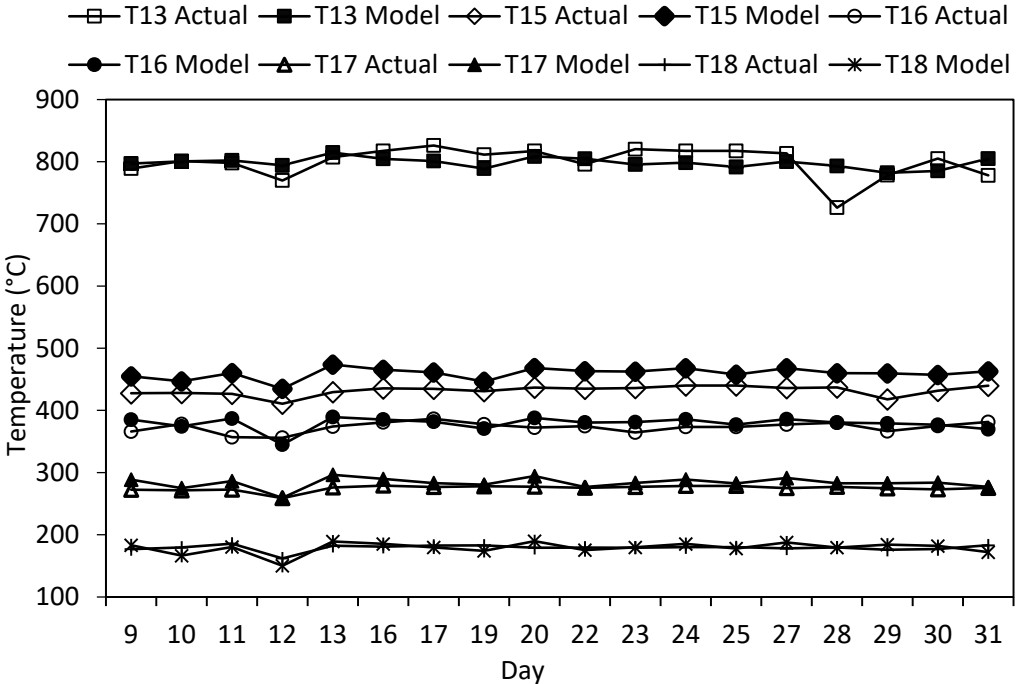

**Figure 3.** Temperatures measured in the boiler.

Results show that the error of the model contrasted with the temperature measured values averaged between 5.8% for T15 and 2.2% for T13. These results show the validity of the model in predicting the performance of the boiler.

### 3.2. Exergy Balance

Figure 4 shows the exergy balance of the boiler. This figure shows that the exergy destruction and loss in the boiler account for 72.2% of the exergy input. The exergy destruction in the combustion chamber and the economizer sums up 49.2% of the exergy input, while the steam production system accounts for 12.8%. Overall, the exergy destruction accounts for 72.2%, and the current operation's boiler efficiency results in 27.8%.

The share of exergy lost and destroyed in the different subsystems in the general operation is highlighted in Figure 5. Overall, 65% of the exergy destruction and loss occurs in the combustion chamber as a result of combustion irreversibilities. Furthermore, the steam production system accounts for 18% of the exergy destruction. In total, the combustion chamber, and the steam production system account for 83% of the exergy destruction in the boiler. Moreover, the superheaters account for 11% of the exergy destruction. The remaining systems combined account for 6% of the exergy destruction and loss.

### 3.3. Optimization of the Steam Boiler

The optimization identifies the Pareto front, as shown in Figure 6. Table 6 shows the operation parameters of the steam boiler obtained with the optimization process and their comparison with the measured parameters during the optimal operation.

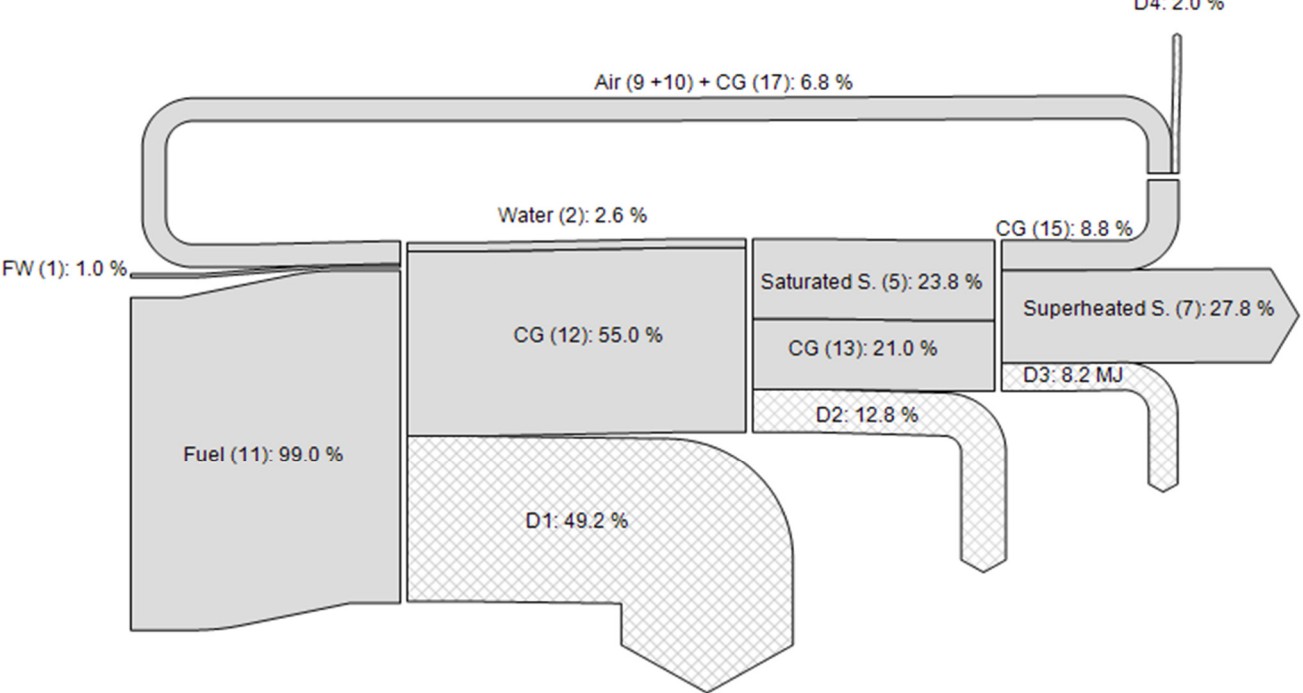

**Figure 4.** Sankey diagram of the steam boiler. FW—Feedwater, CG—Combustion gases, Air— includes primary and pneumatic air, Saturated S.—Saturated steam, Superheated S.—Superheated steam, D—Exergy destroyed.

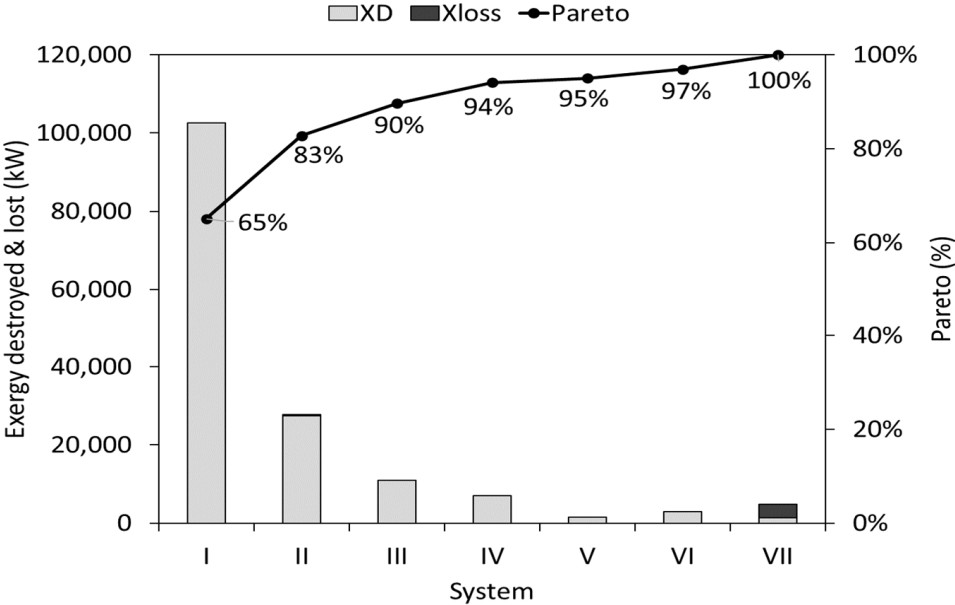

**Figure 5.** Distribution of exergy destruction and losses in the boiler during general operation.

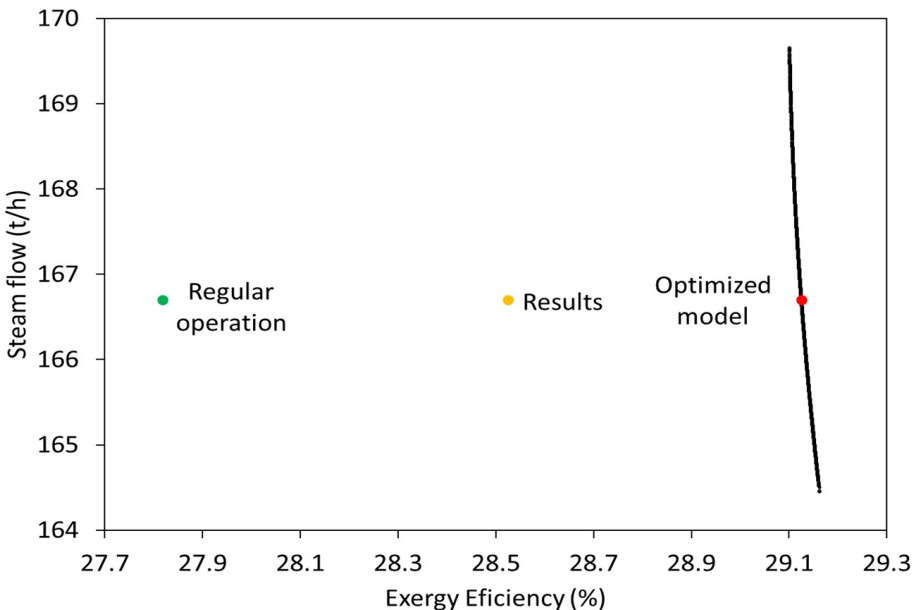

**Figure 6.** Pareto front for steam flow and boiler efficiency.

**Table 6.** Current and optimized operation of the boiler.

| Operation Parameters | Flow | Unit | Acronym | Optimized Operating Point (ModeFRONTIER) | Optimized Operating Point (Implementation) |
|---|---|---|---|---|---|
| Steam flow | 7 | t/h | | 166.7 | 165.1 |
| Steam temperature | 7 | °C | $T_7$ | 506 | 513 |
| Steam pressure | 7 | kPa | $P_7$ | 6691 | 6552 |
| Feedwater flow | 1 | t/h | $F_1$ | 166.7 | 163.5 |
| Feedwater temperature | 1 | °C | $T_1$ | 124.6 | 116.5 |
| Feedwater pressure | 1 | kPa | $P_1$ | 6900 | 6800 |
| Water temperature | 2 | °C | $T_2$ | 182.3 | 185 |
| Primary airflow | 10 | t/h | $F_{10}$ | 177 | 165 |
| Primary air temperature | 10 | °C | $T_{10}$ | 190 | 243 |
| Secondary airflow | 9 | t/h | $F_9$ | 75 | 70 |
| Secondary air temperature | 9 | °C | $T_9$ | 280 | 384 |
| Pneumatic airflow | 8 | t/h | $F_8$ | 21 | 20 |
| Pneumatic air temperature | 8 | °C | $T_8$ | 30 | 30 |
| Input air | 9.1 | °C | $T_{9.1}$ | 30 | 30 |
| | 10.1 | °C | $T_{10.1}$ | 30 | 30 |
| Gas temperature | 12 | °C | $T_{12}$ | 1534 | 1520 |
| | 18 | °C | $T_{18}$ | 175 | 180 |
| | 17 | °C | $T_{17}$ | 273.7 | 277 |

Table 6 shows that, although it is feasible to improve the boiler efficiency, the control over the variables is complex and the improvement is limited. The bagasse flow for the optimized operation point accounts for 1886.6 t/day.

The optimal operating point was implemented during 18 days of boiler operation following different strategies to guarantee the highest performance efficiency:

- To improve the combustion process:
  - ✓ Primary airflow: This flow is set by the boiler operator. Usually, this is the main combustion airflow, which is suboptimal for 100% bagasse.
  - ✓ Secondary airflow: Pneumatic air flow (AFPF): This flow complements the primary air flow to increase combustion efficiency with preheated air.
  - ✓ Pneumatic air flow: Flow is needed to blow bagasse away from the discharge to prevent shorter particle residence time, leading to lower combustion efficiency. However, due to its lower temperature contrasted to the primary and secondary air flows, it must be controlled to the minimum to obtain higher combustion efficiency.

The operational points measured during regular operation and from the optimization are depicted in Figure 6. These points are classified as follows:

- Regular operating configuration from Table 2 (green point in Figure 6).
- Pareto front: nondominated optimal points for different flows (black points in Figure 6).
- Optimal operating point from the model: operational point selected from the Pareto front (red point in Figure 6).
- Result from implementing the optimized operating point (yellow point in Figure 6).

In the sugar plant, using high-efficiency extraction mills results in an average bagasse moisture of 47%. This average value is used for the optimization of the model.

The implementation of multi-objective optimization resulted in a Pareto front including 1765 solutions in the vicinity of the regular operating point of the boiler. The boiler regularly operates, producing 166.7 t/h at an exergy efficiency of 27.8%. The optimal efficiency in the Pareto front was thus selected at this steam flow. The optimal point, in this case, coincides with an exergy efficiency of 29.1%. Implementing the optimal operating configuration obtained from the optimization in the boiler yielded an exergy efficiency of 28.6% (0.8% higher than the regular operation).

Figure 7 shows the fuel consumption and the exergy efficiency for 18 days of implementing the optimized parameters.

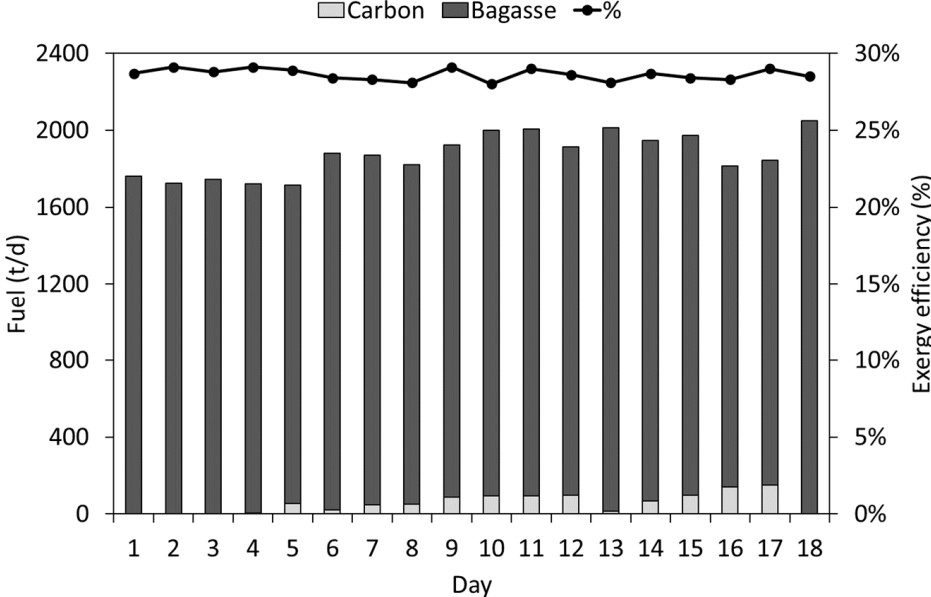

**Figure 7.** Result from the implementation of the optimized parameters.

Overall, the exergy efficiency averaged 28.6% while varying between 28% and 29.1% during the implementation. In this period, the efficiency varied little, showing good control of the boiler operation. The differences with the model are explained since certain properties and characteristics of bagasse, such as chemical composition and moisture, vary with the sugarcane processed in the plant. Other considerations are also used in the mathematical model, which affects its accuracy. These are the main limitation of this study. Future studies are projected to improve these limitations by developing a model considering heat transfer, pressure drops, bagasse combustion, and other processes that will improve the accuracy of the model.

Furthermore, the measured characteristics of bagasse at the plant input need to be integrated into the model to improve the prediction accuracy. Overall, bagasse demand during the test reduced from an average of 1909 t/d to 1887 t/d, with monthly savings of 7471 t. Excess bagasse can be commercialized to paper production plants at 10.6 USD, equivalent to 50,000 COP/t considering an exchange rate of 4700 COP/USD. Bagasse savings account for 7225 USD per year.

## 4. Conclusions

Based on the results obtained for the boiler, the model can be used to forecast and control the fuel demand with an error lower than 6%. In total, the combustion chamber accounts for 65% of the exergy destruction in the boiler, mainly as a result of fuel combustion. Thus, closely monitoring the combustion process is essential in maintaining and improving boiler efficiency. Furthermore, the steam production system accounts for 18% of the exergy destruction, mainly due to the heat and momentum transfer. The multiobjective optimization upgrades the boiler efficiency by 0.8%, which increases plant energy production. Consequently, the bagasse sales for paper and cardboard production can be increased. Thus, this approach improves the economic profit of the plant.

Moreover, these results encourage the expansion of biomass-based electricity systems and the improvement of the energy efficiency of biomass-based power plants under exploitation. Furthermore, this approach can be expanded to other sectors relying on biomass based electricity such as the oil palm industry. Biomass based electricity systems contribute to the stability of electric grids, preventing grid stability and security issues associated with exploiting other renewable energy sources.

**Author Contributions:** Conceptualization, D.L.M., J.R.V.M. and J.A.L.; methodology, A.S.G., E.C.Q. and J.J.C.E.; software, D.L.M. and J.R.V.M.; validation, D.L.M. and J.R.V.M.; formal analysis, A.S.G., E.C.Q. and J.J.C.E.; investigation, D.L.M. and J.R.V.M.; writing—original draft preparation, D.L.M. and J.R.V.M.; writing—review and editing, E.C.Q. and S.H. All authors have read and agreed to the published version of the manuscript.

**Funding:** This research was funded by "Universidad Autónoma de Occidente" grant number (21INTER-387) And The APC was funded by project "Propuesta para el Establecimiento y Formalización del Centro de evaluación PEVI—UAO (code: 21INTER-387)".

**Data Availability Statement:** Data is available within the manuscript.

**Acknowledgments:** The authors thank Kevin Nabor Paredes C. for contributing to this paper. Furthermore, the authors thank the "Universidad Autónoma de Occidente" for its support as a part of the industrial assessment program.

**Conflicts of Interest:** The authors declare no conflict of interest regarding the publication of this paper.

## Nomenclature

| | |
|---|---|
| $c_p$ | Specific heat |
| GHG | Greenhouse gases |
| h | Specific enthalpy |
| KE | Kinetic exergy |
| LHV | Lower heating value |

| | |
|---|---|
| m | Mass flow |
| P | Pressure |
| PE | Potential exergy |
| Q | Heat flow |
| R | Gas constant |
| S | Entropy |
| t | Time |
| T | Temperature |
| TL | Temperature of the heat sink |
| TH | Temperature of the heat source |
| x | Specific exergy |
| X | Exergy flow |
| W | Power |
| V | Volume |
| Greek letters: | |
| φ | Fuel factor |
| η | Exergy efficiency |
| **Subscripts:** | |
| a | Air |
| boiler | Boiler |
| CV | Control volume |
| D | Destroyed |
| f | Fuel |
| g | Gas |
| gen | Generation |
| loss | Loss |
| s | Steam |
| SH | Superheater |
| 0– | Ambient/Reference value |

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
