# Peer review of "Multiobjective Optimization of the Energy Efficiency and the Steam Flow in a Bagasse Boiler"

_sustainability, doi:10.3390/su151411290_

Round 1

Reviewer 1 Report (Previous Reviewer 2)

I have already given my evaluation in my former reports where I recoomended the manuscript for publication. The additional changes made now by tha authors are due to suggestions of other referees.

Author Response

The authors thank the reviewer for its contributions.

Reviewer 2 Report (New Reviewer)

The title and abstract reflect the general idea stated in the text of the article.

The material is presented in an accessible form, presentation and quality of writing is good.

Author Response

The authors thank the reviewer for its contributions.

Reviewer 3 Report (New Reviewer)

The article is well written (english and scientific notation).

The fuels are sugar bagasse and carbon. What is carbon? Mineral char or charcoalor ....? May be carbon is not the corret english word.

Author Response

The reviewer is correct. The word carbon was replaced by coal in the text.

This manuscript is a resubmission of an earlier submission. The following is a list of the peer review reports and author responses from that submission.

Round 1

Reviewer 1 Report

Please see the attached pdf. 

Author Response

Detailed response to reviewers' comments:

Review of the Multi-objective optimization of the energy efficiency and the steam flow in a bagasse boiler titled manuscript.

As I have written in my previous review, the topic is up-to-date, and due to the corrections according to the reviewers comments, the quality of the manuscript is increased, but in my opinion, it has to be improved.

  1. The editing influences of the soundness of the article. So in Page 1, the problems in the nomenclature and subscripts is not acceptable. If I see an article such like this, I will not read more from it.

The editing of the nomenclature was accordingly improved.

  1. There are a lot of i.e. in the written text, which is not so elegant.

The author thanked the reviewer for this remark, the manuscript was reviewed, and only one i.e., remains.

  1. In Page 4 is almost blank. In my opinion Table 2 fits in this space.

The reviewer is correct. Table 2 was moved to page 4.

  1. When you described the Pareto optimization, I do not know which will be the objective functions. Somewhere you should write that f1(x) is this and f2(x) is that, and maybe you have to write about the condition functions and constraints also. This can be the main impact of your article.

Following the reviewer's recommendation, the objective functions and constrains are highlighted and discussed on page 7.

  1. I do not like when the authors do not use subscripts and superscripts. Please correct the signs in Table 5.

The nomenclature in table 5 was accordingly corrected using subscripts.

  1. I still do not understand the presence of Figure 6. Please describe your conclusions in more detail.

Figure 6 was removed from the paper.

Reviewer 2 Report

The authors have re-submitted a consierably improved version of the manuscript. They have picked up all of the sugggestions given in my first review report. To all questions they have given reasonable answers. Therefore, from nmy point of view, I can now recommend publication of the revised version as is in the scinetific jouranl sustainability.

Author Response

The authors have re-submitted a consierably improved version of the manuscript. They have picked up all of the sugggestions given in my first review report. To all questions they have given reasonable answers. Therefore, from nmy point of view, I can now recommend publication of the revised version as is in the scinetific jouranl sustainability.

  1. The authors thank the reviewer for his remarks that helped improve the paper.